# Toxicity of UV Filter Benzophenone-3 in Brine Shrimp Nauplii (*Artemia salina*) and Zebrafish (*Danio rerio*) Embryos

**Melissa I. Ortiz-Román** *, **Ileska M. Casiano-Muñiz** and **Felix R. Román-Velázquez** *

Department of Chemistry, University of Puerto Rico, Mayaguez Campus, Mayaguez, PR 00681, USA; ileska.casiano@upr.edu
* Correspondence: melissa.ortiz10@upr.edu (M.I.O.-R.); felixr.roman@upr.edu (F.R.R.-V.)

**Abstract:** The benzophenone (BP) family, including oxybenzone (BP-3), a prevalent sunscreen ingredient and environmental contaminant, has raised concerns since the year 2005. This study investigated oxybenzone toxicity in zebrafish (*Danio rerio*) eleutheroembryos and brine shrimp (*Artemia salina*) nauplii, focusing on the $LC_{50}$ and developmental impacts. Zebrafish embryos (0.100–1.50 mg/L BP-3, 96 h) and *A. salina* (0.100–5.00 mg/L BP-3, 48 h) were tested with ultrasound-assisted emulsified liquid-phase microextraction (UA-ELPME) used for zebrafish tissue analysis. HPLC-DAD determined BP-3 concentrations (highest: $0.74 \pm 0.13$ mg/L). Although no significant zebrafish embryo mortality or hatching changes occurred, developmental effects were evident. Lethal concentrations were determined (*A. salina* $LC_{50}$ at 24 h = $3.19 \pm 2.02$ mg/L; *D. rerio* embryos $LC_{50}$ at 24 h = $4.19 \pm 3.60$ mg/L), with malformations indicating potential teratogenic effects. *A. salina* displayed intestinal tract alterations and *D. rerio* embryos exhibited pericardial edema and spinal deformities. These findings highlight oxybenzone's environmental risks, posing threats to species and ecosystem health.

**Keywords:** UV filter; benzophenones; oxybenzone; emerging contaminant; zebrafish embryos; lethal concentration; brine shrimp; toxicity; malformations

## 1. Introduction

In the 1950s, benzophenones (BP) and their derivatives were introduced into sunscreens [1]. Oxybenzone (2-hydroxy-4-methoxybenzophenone, BP-3), which is derived from benzophenone, is a common organic ultraviolet (UV) filter used in sunscreens and other personal care products (nail polish, lotions, and lipsticks). The maximum concentration allowed is 6% (*w/v*) in the United States of America (USA) [2]. The amount of BP-3 used in USA sunscreens is greater than all other benzophenones combined; it was found in 68% of the 201 sunscreens assessed [3]. Oxybenzone is one of the most frequently detected UV filters in surface waters and wastewater. Since 2005, it has been listed as an emerging contaminant owing to its worldwide occurrence [4].

Environmental concentrations of BP-3 in natural waters range between 0.7 and 7.8 μg/L [5]. In the USA, an incredibly high concentration of 1.395 mg/L was detected along Trunk Bay in the Virgin Islands [6]. In adult zebrafish (*Danio rerio*) and eleutheroembryos, concentrations raging 2.4–312 μg/L led to the downregulation of enzymes involved in steroidogenesis and hormonal pathways. Low concentrations of BP-3 exhibit similar multiple hormonal activities at the transcriptional level in two different life stages of zebrafish [7]. Exposure to BP-3 causes mortality, unsuccessful hatching, and various malformations in zebrafish embryos [8]. Studies have demonstrated the bioaccumulation of UV filters in wildlife (e.g., fish) owing to the lipophilic nature of these chemicals [9,10]. A summary of the effects of BP-3 reported by other authors is presented in Table 1.

Zebrafish are currently considered excellent model organisms in various biomedical fields, including developmental toxicology [11]. Low cost, easy maintenance, transparent

embryos, easy manipulation, high fecundity, and rapid embryonic development make zebrafish an attractive model for in vivo assays. Guidelines from the Organization of Economic Cooperation and Development (OECD) recommend zebrafish for aquatic toxicity testing [12–14]. Zebrafish undergo rapid development, facilitating the evaluation of various toxicological endpoints within a few days post-fertilization. Since the larval and early fry stages are the most sensitive for assessing the developmental effects of xenobiotics, OECD Guideline 212 [14] and OECD test Guideline 236 [15] focus on toxicity during early fish development. Since the fish embryo toxicity test (FET) has become a potential alternative to acute fish toxicity testing [16], zebrafish embryos (ZFE) are gaining popularity in hazard assessment.

**Table 1.** Previous studies of BP-3 effects.

| Organism | Title/Authors | BP-3 Concentrations | Conclusions |
|---|---|---|---|
| Zebrafish Embryos (*Danio rerio*) | Hormonal activity, cytotoxicity, and developmental toxicity of UV filters [8] | Concentrations used: 1.00 mg/L–25.00 mg/L | • $LC_{50}$ value (15.93 mg/L) was found for the zebrafish embryos after 96 h postfertilization (hpf) of exposure. <br> • Decreased the number of the hatched embryos. <br> • Caused pericardial and yolk sac oedema in the embryos. <br> • Deformation of the tail |
| Zebrafish (*Danio rerio*) | Effects of the UV filter benzophenone-3 (oxybenzone) at low concentrations in zebrafish (*Danio rerio*) [7] | Concentrations used: 0.010, 0.200, and 0.600 mg/L. | • Eleutheroembryos displayed normal swimming behavior by the end of the 120 h exposure. <br> • BP-3 is metabolized to BP-1 in adult zebrafish, but not in eleutheroembryos. |
| Tilapia fillets (*Oreochromis urolepis hornorum*) | Methodology for Analysis of UV filters in Tilapia using off-line MSPD followed by On-line SPE-LC-UV [17] | Concentrations used: 500 μg/mL with five UV filters. | • The concentration of oxybenzone at 24 h was 0.540 and 1.350 mg/L and increased to 1.250 and 2.900 mg/L after 72 h of exposure. |
| *Artemia salina* (nauplii instar II/III) | Effect of 10 UV filters on the brine shrimp *Artemia Salina* and the marine microalga *Tetraselmis* sp. [18] | Concentrations used: 20 ng/L and 2 mg/L. | • No toxicity was observed for BP-3, even at the highest concentration. |
| *Artemia franciscana* and *Daphnia magna* | Acute toxicity assessment of nine organic UV filters using a set of biotests [19] | Concentrations used: 10, 12.5 mg/L | • *Artemia franciscana* exhibited an $LC_{50}$ of 5.27 mg/L after a 48-h exposure to BP-3. <br> • *Daphnia magna* demonstrated an $EC_{50}$ of 3.25 mg/L after a 48-h exposure. |
| Zebrafish (*Danio rerio*) | Lifetime exposure to benzophenone-3 at an environmentally relevant concentration leads to female–biased social behavior and cognition deficits in zebrafish [20] | Concentration used: 10 μg/L | • Continuous exposure from 6 h post fertilization to adulthood at 5 months induced social behavior and learning/memory deficits are correlated to the cell loss in the telencephalon region of the zebrafish brain. |

**Table 1.** *Cont.*

| Organism | Title/Authors | BP-3 Concentrations | Conclusions |
|---|---|---|---|
| Zebrafish (*Danio rerio*) Embryos | Embryonic benzophenone-3 exposure inhibited fertility in later-life female zebrafish and altered developmental morphology in offspring embryos [21] | Concentrations used: 0, 1, 10, 100 µg/L | • Inhibited the development and maturation of ovaries in later-life female zebrafish, reduced egg production and delayed hatching rate. |
| Zebrafish (*Danio rerio*) Embryos | Comparison of developmental toxicity of benzophenone-3 and its metabolite benzophenone-8 in zebrafish [22] | Concentrations used: 1, 30, 300, 3000 µg/L | • Caused behavioral abnormalities of zebrafish larvae.<br>• Altered the metabolism by pantothenate and CoA biosynthesis pathway.<br>• Altered neuroactive ligand-receptor interaction pathway and FoxO signaling pathway.<br>• Changed the cofactor and vitamin metabolism of zebrafish larvae. |
| Adult zebrafish (*Danio rerio*) | Effects of oxybenzone on zebrafish behavior and cognition [23] | Concentrations used: 10, 100 and 1000 µg/L | • Affects perception, increases risk-taking, impairs proper aggressive response, and jeopardizes the animal's ability to retain information. |
| Zebrafish embryos (*Danio rerio*) | Effects of Low Concentration Benzophenone-3 Exposure on the Sex Ratio and Offspring Development of Zebrafish (*Danio rerio*) [24] | Concentrations used: 0, 0.056, 2.3, and 38 µg/L | • Concentration of 38 µg/L can lead to a significant skew in the sex ratio in zebrafish at 42 dpf.<br>• Up-regulate vtg2 at 42 dpf.<br>• Endocrine disruptive effects on zebrafish. |
| Zebrafish (*Danio rerio*) larvae | Toxic effects and transcriptome analyses of zebrafish (*Danio rerio*) larvae exposed to benzophenones [25] | Concentrations used: 0.91 to 4.56 mg/L | • Inhibited the hatching of and caused larval deformity.<br>• The $LC_{50}$ value of larvae (50 hpf) 9.88 mg/L at 25 h and 4.15 mg/L at 96 h.<br>• Endocrine disruptor, affects the expression of cytochrome P450.<br>• Affected estradiol biosynthesis and sex differentiation.<br>• Increase CYP1A and CYP1B expressions and might mediate the oxidative metabolism of estrogen. |
| Zebrafish (*Danio rerio*) | Environmental relevant concentrations of benzophenone-3 induced developmental neurotoxicity in zebrafish [26] | Concentrations used: 0.04 µM | • Caused effects including altered motor and social behaviors in zebrafish larvae.<br>• Decreases tactile response at 27 hpf.<br>• Neurotoxic to developing embryos in zebrafish.<br>• Affected cell proliferation and apoptosis in the larval head region. |

**Table 1.** *Cont.*

| Organism | Title/Authors | BP-3 Concentrations | Conclusions |
|---|---|---|---|
| Zebrafish (*Danio rerio*) embryo | Thyroid Hormone-Disrupting Potentials of Major Benzophenones in Two Cell Lines (GH3 and FRTL-5) and Embryo-Larval Zebrafish [27] | Concentrations used: 0, 32, 100, and 320 µg/L | <ul><li>Showed a significant decrease of T3 levels, but not T4 levels.</li><li>Potential thyroid hormone disruptor.</li><li>BP-3 significantly up-regulated the tg, dio1, and ugt1ab genes</li></ul> |
| Zebrafish (*Danio rerio*) | Endocrine disrupting effect of the ultraviolet filter benzophenone-3 in zebrafish, *Danio rerio* [28] | The concentrations used: 100, 32, 500 mg/L | <ul><li>Caused an adverse endocrine-disrupting effect in developing zebra fish by showing phenotypic sex toward females.</li><li>Interferes with the maturation stages of the gonads.</li></ul> |

*Artemia salina* is a species of zooplankton that plays an important role in the energy flow of the food chain in the marine environment [29] and is used to feed larval fish. The brine shrimp assay is quick, affordable, and easy to use, making it convenient. When stored dry, *A. salina* eggs are easily accessible, reasonably priced, and viable for many years [30]. *A. salina* is a convenient test organism for toxicity studies [31]. *A. salina* functions as a commonly employed model organism in toxicological assessments, offering flexibility in the selection of various parameters including hatching, mortality, swimming behavior, and morphology [32]. Therefore, brine shrimp has been chosen as an alternative to in vitro cell culture tests [30].

This study seeks to contribute to the development of novel methods for assessing the toxicity of pollutants continually introduced into water bodies through human activities. An additional objective was to gain deeper insight into the potential toxicological effects of exposure to varying concentrations of BP-3 on *Danio rerio* embryos and *A. salina*. Identifying and documenting xenobiotics in water, understanding their sources, and assessing their potential effects are crucial. Increasing awareness of these environmental threats is key to encouraging policymaking. Continued research on the effects of xenobiotics on aquatic organisms is imperative to provide valuable insights into their potential human impacts. The outcomes could formulate new regulations, including potential bans on oxybenzone in personal care products or the establishment of broader environmental policies.

## 2. Materials and Methods

### 2.1. Preparation of Xenobiotic for Artemia salina Nauplii Assay

A stock solution of concentrated (200.0 mg/L) oxybenzone (CAS: 131-57-7; Sigma-Aldrich, St. Louis, MO, USA) with HPLC-grade methanol (CAS: 67-56-1; Sigma-Aldrich, St. Louis, MO, USA) was prepared. It was placed ultrasonically (Bransonic®, model 3510R-MT, Branson Ultrasonics Corp., Danbury, CT, USA) for 15 min. Six concentrations were prepared for the treatments: 0.100, 0.300, 0.500, 1.00, 3.00, and 5.00 mg/L of BP-3. The treatment solution was diluted with artificial seawater (prepared in deionized water dissolving Instant Ocean Sea salt (60 g/L). They were then placed on ultrasound (Bransonic®, model 3510R-MT) for 30 min.

### 2.2. Artemia salina Nauplii Toxicity Assay

*A. salina* was obtained from commercially dried encysted eggs. The cysts were incubated in artificial seawater for 48 h at 25 ± 1 °C, consistently illuminated by a 60 W lamp, and aerated using an air pump. Subsequently, the hatched *A. salina* were carefully transferred using pasteur pipettes into glass test tubes containing 5 mL of xenobiotic (BP-3, diluted in synthetic seawater) for the experimental assay. Each test tube contained ten

organisms. Mortality assessments were conducted after 24 h and 48 h of exposure. The concentrations used were 0.10, 0.30, 0.50, 1.0, 3.0, and 5.0 mg/L of BP-3. Three control experiments were performed: synthetic seawater alone, synthetic seawater with 1.5% methanol (corresponds to the methanol percentage in the 3.0 mg/L exposure group), and synthetic seawater with 2.5% methanol (corresponds to the methanol percentage in the 5.0 mg/L exposure group). These controls were implemented to systematically rule out any potential toxicity resulting from the methanol sourced from the BP-3 stock. Only these percentages of methanol were analyzed, as concentrations greater than 1% can be harmful to organisms. The treatment groups and controls were analyzed in triplicate. After the experiment, *A. salina* was euthanized using 70% ethanol (CAS:64-17-5; Sigma-Aldrich, St. Louis, MO, USA).

### 2.3. Fish Maintenance and Breeding: Zebrafish

Mature wild-type zebrafish obtained from Caribe Fisheries Inc. (fish farm) in Lajas, Puerto Rico were housed on a ZebTEC Benchtop zebrafish housing system (Tecniplast, Buguggiate (VA) Italy) with mechanical and activated carbon filtration and sterilization (ultraviolet 40 W disinfection) at an automatized 12/12 h light/dark cycle. The water temperature was set to 27 °C ± 1 °C, and the room temperature was set to 24 ± 1 °C. Reverse osmosis (RO) + UV water, commercial sea salts (Instant Ocean® Sea Salt, St. Blacksburg, VA, USA), and $NaHCO_3$ (Sigma-Aldrich, St. Louis, MO, USA)were added to maintain pH and conductivity values of 7.0 ± 0.5 and 1000 ± 100 μS, respectively. This water is referred to as embryo water. Properties such as the pH, conductivity, system water temperature, room temperature, and system volume were checked and recorded daily. Adult fish were fed twice daily on a Zeigler Adult Zebrafish Diet.

### 2.4. Reproduction Process

Sexually mature zebrafish were selected and placed in crossing tanks (transparent plastic fish tanks) with a mesh to prevent the fish from eating their eggs. The female-to-male ratio was no more than 1:2. Crossing tanks were placed overnight on a rack. Embryos were collected in a glass Petri dish and rinsed with embryo water the following morning.

### 2.5. Preparation of Xenobiotic for a Zebrafish Embryo Assay

A concentrated stock solution of oxybenzone (CAS: 131-57-7; Sigma-Aldrich, St. Louis, MO, USA) at 200.0 mg/L was prepared using HPLC-grade methanol (CAS: 67-56-1; Sigma-Aldrich, St. Louis, MO, USA). It was placed ultrasonically (Bransonic®, model 3510R-MT, Branson Ultrasonics Corp., Danbury, CT, USA) for 15 min. Five concentrations were prepared for the treatments: 0.100, 0.300, 0.500, 1.00, and 1.50 mg/L. The treatment solution was diluted with embryo water (without having embryos) (RO + UV water with pH 7.0 ± 0.5 and conductivity 1000 ± 100 μS). They were then placed on ultrasound (Bransonic®, model 3510R-MT, Branson Ultrasonics Corp., Danbury, CT, USA) for 30 min. The percentage of methanol in the treatments did not reach 1% [33]. These concentrations were based on the environmental concentrations found in other studies and their low solubility in water (6.0 mg/L at 25 °C [34]). The diluent was embryo water to simulate the natural environment of zebrafish embryos as well as to avoid other possible variables that could affect the embryos.

### 2.6. Fish Embryo Acute Toxicity Test

The test method was based on OECD guideline 236 [15]. Fertilized embryos (2–4 h post-fertilization) were collected and washed with embryo water. Embryos were examined for vitality using a stereomicroscope (AmScope SM-2TZ-LED; Irvine, CA, USA.) and were randomly distributed to the experimental groups (Scheme 1). The experimental design consisted of six groups: one control group and five experimental groups. Embryos were transferred to glass Petri dishes containing oxybenzone treatment solution. Thirty embryos were placed in each dish and three replicates were evaluated per treatment. The

control treatment contained only the embryo water without oxybenzone. Embryos were continuously exposed at 96 hpf at 28 ± 1 °C in Benchmark's My Temp™ Mini Digital Incubators (H2200-H; Sayreville, NJ, USA) in glass Petri dishes under a photoperiod of 12:12 h light/dark and were not fed during exposure. After 24, 48, 72, and 96 h, the appearance, mortality, development, and abnormal behavior were visually inspected and recorded using a stereomicroscope (AmScope SM-2TZ-LED) and a Trinocular Compound Microscope (OMAX M837SL; Kent, WA, USA) with an 18.0 MP Digital camera (OMAX A35180U3). Dead embryos or eleutheroembryos were removed and discarded. After the experiment, the eleutheroembryos were immersed in ice baths (0 °C) for 30 min to euthanize them. Subsequently, they were transferred to 1.5 mL centrifuge tubes and stored in a freezer at −20 °C for subsequent procedures. No other reagents were used to euthanize the eleutheroembryos so as not to affect the subsequent analyses. Three rinses with embryo water were performed to remove any external excess, removing as much water as possible before weighing and freezing.

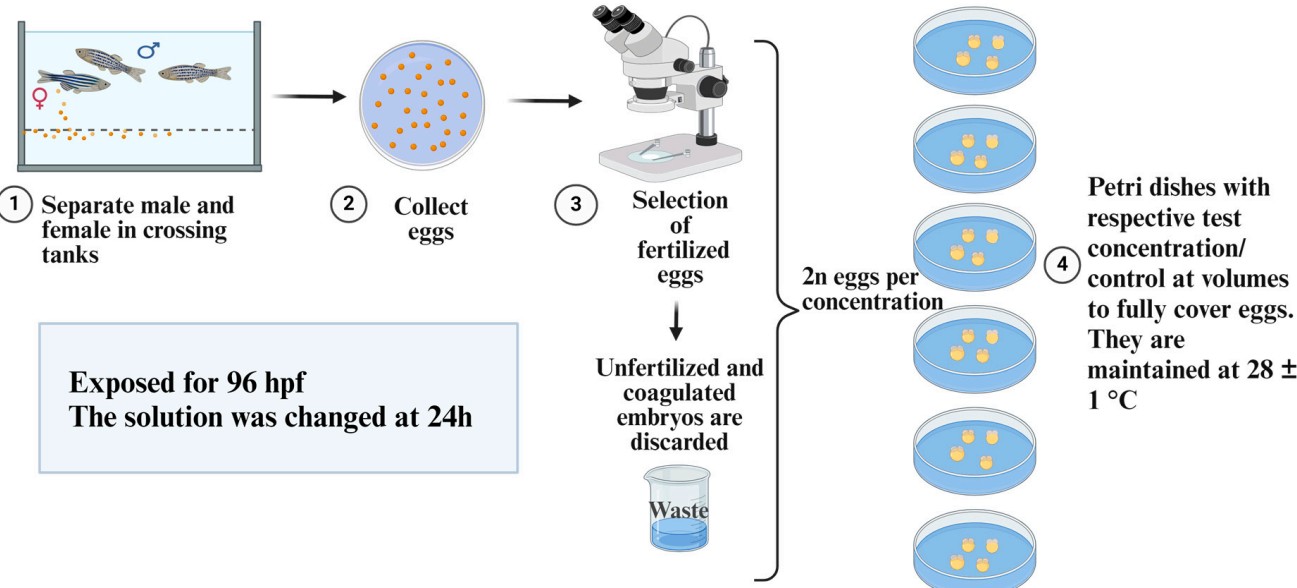

**Every 24 hours the appearance, mortality, development, and abnormal behavior were visually inspected and recorded daily and dead embryos were removed.**

**Scheme 1.** Scheme of the zebrafish embryo toxicity test procedure. Modified image [15]. Created using BioRender.com (accessed on 27 December 2023). Experimental settings for egg production and collection, pre-exposure immediately after fertilization in glass Petri dishes, selection of fertilized eggs with a stereomicroscope, and distribution of fertilized eggs into Petri dishes prepared with their respective test concentrations.

### 2.7. Ultrasound-Assisted Emulsified Liquid Phase Microextraction

An ultrasound-assisted emulsified liquid-phase microextraction (UA-ELPME) method was used to analyze zebrafish larval tissues (Scheme 2). The larval tissue was homogenized with a glass rod in a 1.5 mL centrifuge tube with 300 μL of deionized water (DI water). The extraction procedure involved the addition of 500 μL hexane mixed isomers 98+% (CAS: 92112-69-1, Alfa Aesar, Ward Hill, MA, USA), pre-filtered with a 0.45-micron filter, to the sample in a 1.5 mL centrifuge tube. Subsequently, ultrasonication was applied for 2 h to facilitate extraction, followed by centrifugation at 8000 rpm for 5 min to separate the phases. The upper layer obtained from the previous extraction step was carefully transferred to a new 1.5 mL centrifuge tube. To ensure final purification, the sample was centrifuged at room temperature at 8000 rpm for 5 min. A 100 μL sample was transferred to an HPLC vial

and subjected to evaporation, followed by redissolution to achieve a final volume of 500 μL in HPLC-grade methanol, which had been filtered through a 0.45-micron filter.

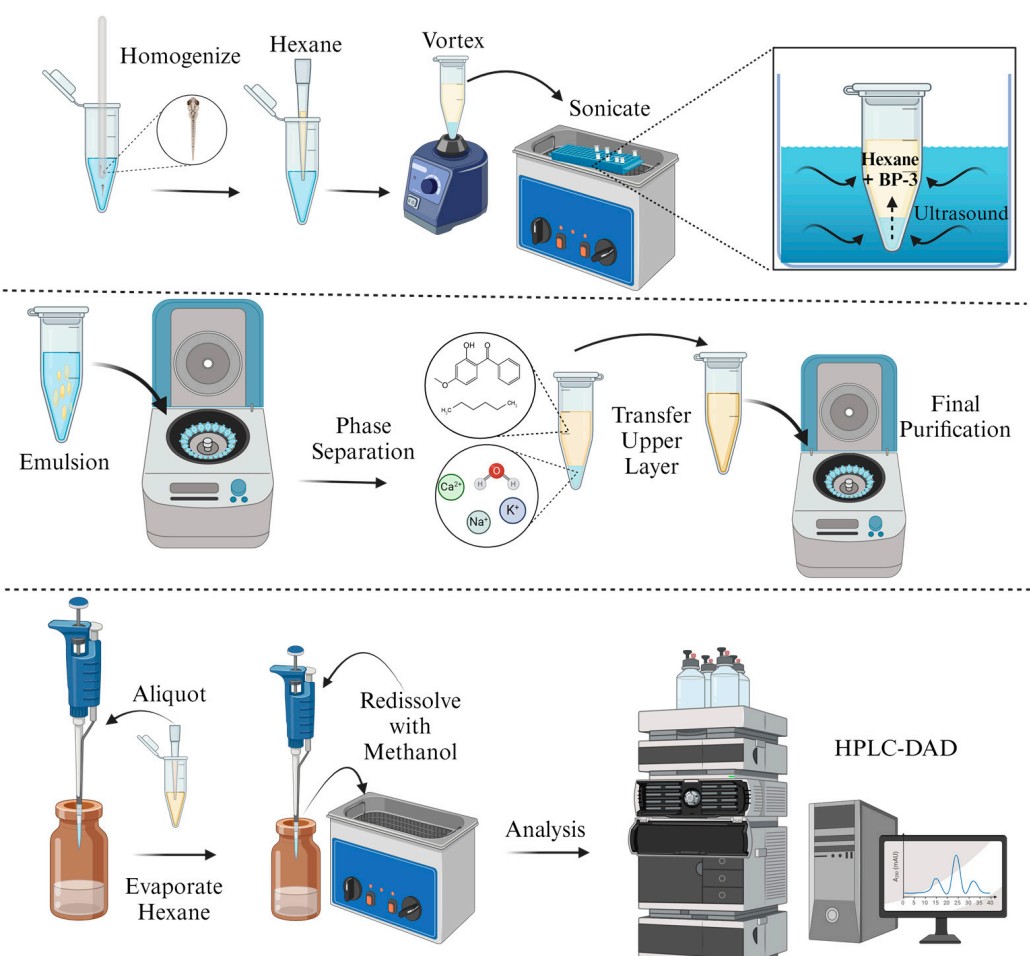

**Scheme 2.** Ultrasound-assisted emulsified liquid-phase microextraction. Created with BioRender.com (accessed on 27 December 2023).

### 2.8. Quantification of Oxybenzone

An HPLC system (Agilent 1200 Series) equipped with an automatic injector and diode array detector (DAD) was used for sample analysis. The HPLC operating conditions were as follows: separations were performed on a C18 column (Phenomenex; Madrid Avenue Torrance, CA, USA) with a length of 150 × 4.60 mm, 5 μm. Methanol (HPLC grade) and DI water (90:10 proportion) were continuously vacuum degassed for the mobile phase. The column temperature was maintained at 25 °C. Chromatography was performed at a flow rate of 1.0 mL/min. The detection wavelength was 289 nm. Agilent ChemStation software (Rev. B.04.02 (96)) was used for the LC 3D systems (Agilent Technologies, 2001–2009; Santa Clara, CA, USA).

### 2.9. Statistical Analysis

All statistical analyses were performed using Minitab Statistical Software version 21.1 (64-bit). Data are expressed as mean ± standard deviation (SD). Differences were determined by one-way analysis of variance (ANOVA) with a Dunnett, Tukey, Fisher LSD method, and the significance was set at $p < 0.05$.

## 3. Results and Discussion

### 3.1. Artemia salina Nauplii Toxicity

The Reed–Muench [35] and Probit [36] methods were used to calculate the lethal concentration. This method relies on cumulative data of deaths and survivors to estimate the $LC_{50}$ value. This involves calculating logarithmic concentrations, as well as the number of deaths and survivors, cumulative deaths, cumulative survivors, cumulative total, mortality rate, survival percentage, and mortality percentage. These calculated data are then graphed to estimated values of lethal concentration. Probit analysis, as outlined by [36], is a parametric method involving linear regression applied to transformed toxicity data. It is particularly well-suited for characterizing binomial response variables, commonly observed in dose-response experiments. This analytical approach involves transforming the data to fit a linear model, upon which regression analysis is conducted, resulting in estimates of the $LC_{50}$.

Confidence in the test results was ensured by the control samples, which exhibited mortality rates of below 10%. The average $LC_{50}$ for *A. salina* in oxybenzone after 24 h was $3.22 \pm 0.04$ mg/L and for 48 h, $1.25 \pm 0.81$ mg/L. The average values of $LC_{10}$, $LC_{20}$, $LC_{30}$, and $LC_{50}$ of 24 and 48 h for *A. salina* exposed to oxybenzone are given in Table 2. The results obtained for $LC_{50}$ can be compared with those of *A. franciscana*, where the $LC_{50}$ value at 48 h was 5.27 mg/L [19], surpassing our findings at both 24 and 48 h of exposure in *A. salina*. This observation suggests that *A. franciscana* may exhibit increased resistance to xenobiotics. Additionally, another study reported higher $LC_{50}$ values for certain xenobiotics in *A. salina* than in *A. franciscana* [37]. On the other hand, [18] could not report the $LC_{50}$ of *A. salina* with the concentrations studied, nor did they find toxicity when exposing them to BP-3 (0.02–2000 µg/L) [18].

**Table 2.** Comparison of lethal concentrations in *Artemia salina* and *Danio rerio*. at 24 h and 48 h. To estimate the lethal concentration, the results of the Reed and Muench method were used. The average is presented with the standard deviation. (*A. salina*, $n = 3$; *D. rerio*, $n = 4$).

| Organism | | | LC$_{10}$ (mg/L) | LC$_{20}$ (mg/L) | LC$_{30}$ (mg/L) | LC$_{50}$ (mg/L) |
|---|---|---|---|---|---|---|
| *Artemia salina* | 24 h | Graph estimation | <0.100 | $0.417 \pm 0.239$ | $1.881 \pm 0.802$ | $3.254 \pm 0.611$ |
| | | Probit | $0.088 \pm 0.034$ | $0.277 \pm 0.160$ | $0.734 \pm 0.394$ | $3.191 \pm 2.023$ |
| | | Average | $0.088 \pm 0.0340$ | $0.347 \pm 0.099$ | $1.308 \pm 0.811$ | $3.223 \pm 0.044$ |
| | 48 h | Graph estimation | ---- | $0.300 \pm 0$ | $0.427 \pm 0.154$ | $1.829 \pm 0.641$ |
| | | Probit | $0.112 \pm 0.049$ | $0.188 \pm 0.068$ | $0.301 \pm 0.065$ | $0.677 \pm 0.044$ |
| | | Average | $0.112 \pm 0.049$ | $0.244 \pm 0.079$ | $0.364 \pm 0.089$ | $1.253 \pm 0.815$ |
| *Danio rerio* embryo | 24 h | Probit | $0.581 \pm 3.595$ | $1.016 \pm 0.624$ | $1.773 \pm 1.238$ | $4.193 \pm 3.596$ |

The control group only had synthetic seawater and showed no deaths (Figure 1). When comparing the control group treated with 1.5% methanol solvent to the group exposed to a concentration of 3.0 mg/L of BP-3, no statistically significant differences ($p > 0.05$) were observed in the mortality percentage at 24 or 48 h. Conversely, the control group treated with 2.5% methanol solvent exhibited significant differences ($p < 0.05$) compared to the group exposed to a concentration of 5.0 mg/L of BP-3 at both time points. These differences occurred from the considerably higher mortality percentage observed in the 5.0 mg/L group compared to its respective control. This finding confirms that the methanol percentage in the BP-3 solution, at concentrations of 3.00 and 5.00 mg/L, did not influence the mortality rate. However, exposing *A. salina* to various concentrations of BP-3 resulted in noticeable behavioral changes after 48 h. At higher concentrations (1.5, 3.0, and 5.0 mg/L), organisms displayed erratic swimming patterns characterized by rapid and inconsistent movements, contrasting with the regular movement patterns observed in the non-exposed group.

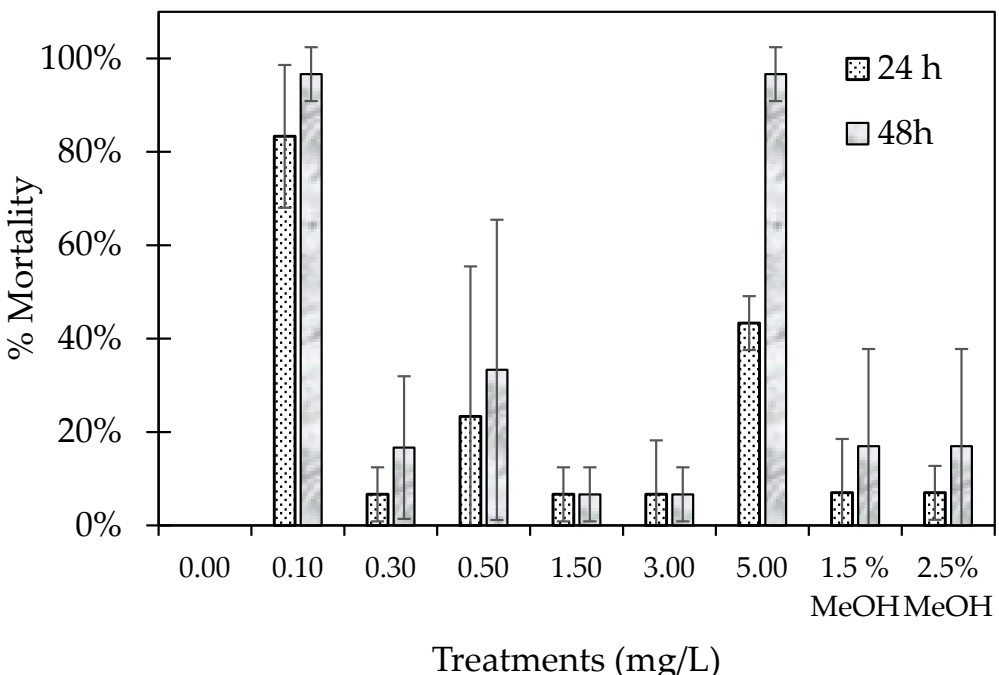

**Figure 1.** % Mortality *A. salina* at 24 and 48 h of exposure. Significant differences were found between treatments at both time points ($p < 0.05$). The average is presented with the standard deviation ($n = 3$).

Following exposure to BP-3 for 48 h, morphological assessments were conducted on *Artemia salina* to evaluate potential changes. Microscopic examination revealed physical malformations in the intestines of brine shrimp exposed to concentrations of 0.300 and 0.500 mg/L of BP-3. Specifically, at the concentration of 0.300 mg/L, a curvature in the intestine near the anus was observed. At the concentration of 0.500 mg/L, it was noted that the digestive tube exhibited irregularity, being attached to one side of the body. These morphological alterations could compromise the proper functioning of the digestive system and potentially lead to the organism's demise in the short term. No antennae or limb abnormalities were observed at any point. No morphological changes were observed in the brine shrimp specimens in the solvent control group.

It is important to highlight that elevated concentrations of BP-3 have been identified in seawater [6]. Consequently, we posit that regions serving as habitats for these brine shrimps, particularly those frequented by tourists, could have considerable effects on their populations. These effects manifest not only in increased mortality rates but also in notable morphological changes, as identified in our findings. This population decline can alter the food chain, thereby affecting other populations. Furthermore, these findings suggest a potential impact on the entire ecosystem, affecting not only brine shrimp populations, but also other organisms coexisting in the same environment.

### 3.2. Fish Embryo Acute Toxicity

The results of the Reed–Muench method [35] and the Probit method [36] were used to calculate the lethal concentrations. The control groups, which were only exposed to embryonic water, showed mortality rates of less than 10%. This demonstrates the strength and trustworthiness of the results obtained in accordance with the OECD guideline. The $LC_{50}$ value could not be determined experimentally at the concentrations used in this study but could be estimated. The $LC_{10}$, $LC_{20}$, $LC_{30}$, and $LC_{50}$ values were estimated using Probit (Table 2). The $LC_{50}$ at 96 h reported by Balázs et al. was 15.93 mg/L for zebrafish larvae exposed to BP-3 at 96 hpf [8]. The concentration mentioned is highly unlikely to be attained in natural water bodies due to the low solubility of oxybenzone in water. The estimated results for lethal concentrations of 10% and 20% of the population are deeply concerning, particularly considering that comparable concentrations have been measured

in isolated areas of aquatic ecosystems. This suggests that in these specific locations, we could potentially witness losses of 10% and 20% of fish embryo populations, which could significantly disrupt the delicate balance of the food web.

Our findings suggest that BP-3 may not exert a notable impact on the mortality of zebrafish embryos during a 96-h exposure under the investigated conditions. However, a significant effect on development was observed. Similarly, another study did not identify a correlation between BP-3 and mortality in zebrafish embryos [7]. Nonetheless, [7] found that even at low concentrations, BP-3 prompts alterations in genes implicated in steroidogenesis and hormonal pathways in zebrafish at varying developmental stages, leading to adverse impacts on the endocrine system. According to our results, exposure to concentrations exceeding 0.100 mg/L of BP-3 could harm these fish, as well as other organisms. No significant differences were found in the hatching rate of zebrafish embryos exposed to BP-3 at concentrations ranging between 0.10 and 1.50 mg/L for 96 h. Hatching rates were calculated cumulatively. Under different experimental conditions, other studies have reported a decrease in the number of hatched embryos attributed to BP-3 [8,38,39]. These differences may stem from various factors or variables employed by each author in their respective research. [8] utilized DMSO as a stock diluent with a conductivity of 525 μS/cm in its medium, which could alter the behavior and interactions of BP-3 during treatment. Conversely, in the study by [38], the same conductivity was maintained in the medium, but DMSO was used as a diluent, leading to their finding that BP-3 can exacerbate oxidative stress associated with fish.

### 3.3. Teratogenic Effects

Morphological changes during embryonic development were observed in all BP-3 treatments. No morphological changes were observed in the control group. The groups that presented a higher percentage of physical malformations were at concentrations of 0.30 and 0.50 mg/L, with 9% malformation (see Figure S1). The physical malformations found in embryos and larvae were severe yolk deformation (SYD), lack of pigmentation (LP), reduced eye size (RES), abnormal body shape (ABS), non-development of eyes (NDE), small head (SH), craniofacial malformation (CFM), barely detached tail (BDT), pericardial edema (PE), short tail (ST), spinal deformities (SD), scoliosis (S), and yolk sac edema (YSE) (Figures 2 and 3). The physical malformations that were most frequently observed in the groups exposed to BP-3 included several PE and SD. Under alternative experimental conditions, Balázs et al. reported that BP-3 induced PE and YSE, as well as tail deformation in zebrafish embryos, particularly at concentrations exceeding those typically encountered in natural waters [8]. Several morphological changes identified following exposure to BP-3 were observed by Incardona et al. (2004) in zebrafish embryos exposed to Polycyclic Aromatic Hydrocarbons (PAHs) [40]. Exposure to fluorene, dibenzothiophene, and phenanthrene, which contain a benzene ring, leads to tail deformation in zebrafish embryos. Additionally, embryos exposed to dibenzothiophene and phenanthrene exhibited severe PE. Comparing our results with those reported by Incardona et al. (2004), it can be suggested that BP-3 exhibits toxicity in the morphology of zebrafish larvae akin to PAHs that possess a three-ring structure. CFM is correlated with the degree of PE [40].

Based on the findings of numerous studies, there are two possible explanations for the observed morphological changes. One such explanation could be the oxidative stress induced by BP-3. Downs (2016) found that BP-3 can induce oxidative stress in corals [6]. Additionally, Zhang demonstrated that ultraviolet light can exacerbate oxidative stress associated with exposure to benzophenone-class ultraviolet filters in fish [38]. Another study by Liu et al. (2015) reported the inhibition of catalase and SOD enzyme activities in the livers of fish (*Carassius auratus*) when exposed to benzophenone or oxybenzone at nominal concentrations of 0.5 or 5 mg/L [41]. Oxidative stress occurs when excess reactive oxygen species (ROS) overwhelm the antioxidant defense system in organisms, leading to lipid peroxidation, protein carbonylation, and DNA damage. Maintaining a balance between oxygen and ROS levels is crucial for normal embryo development. However,

evidence suggests that an altered balance, particularly excessive ROS formation, can result in a wide spectrum of malformations [42].

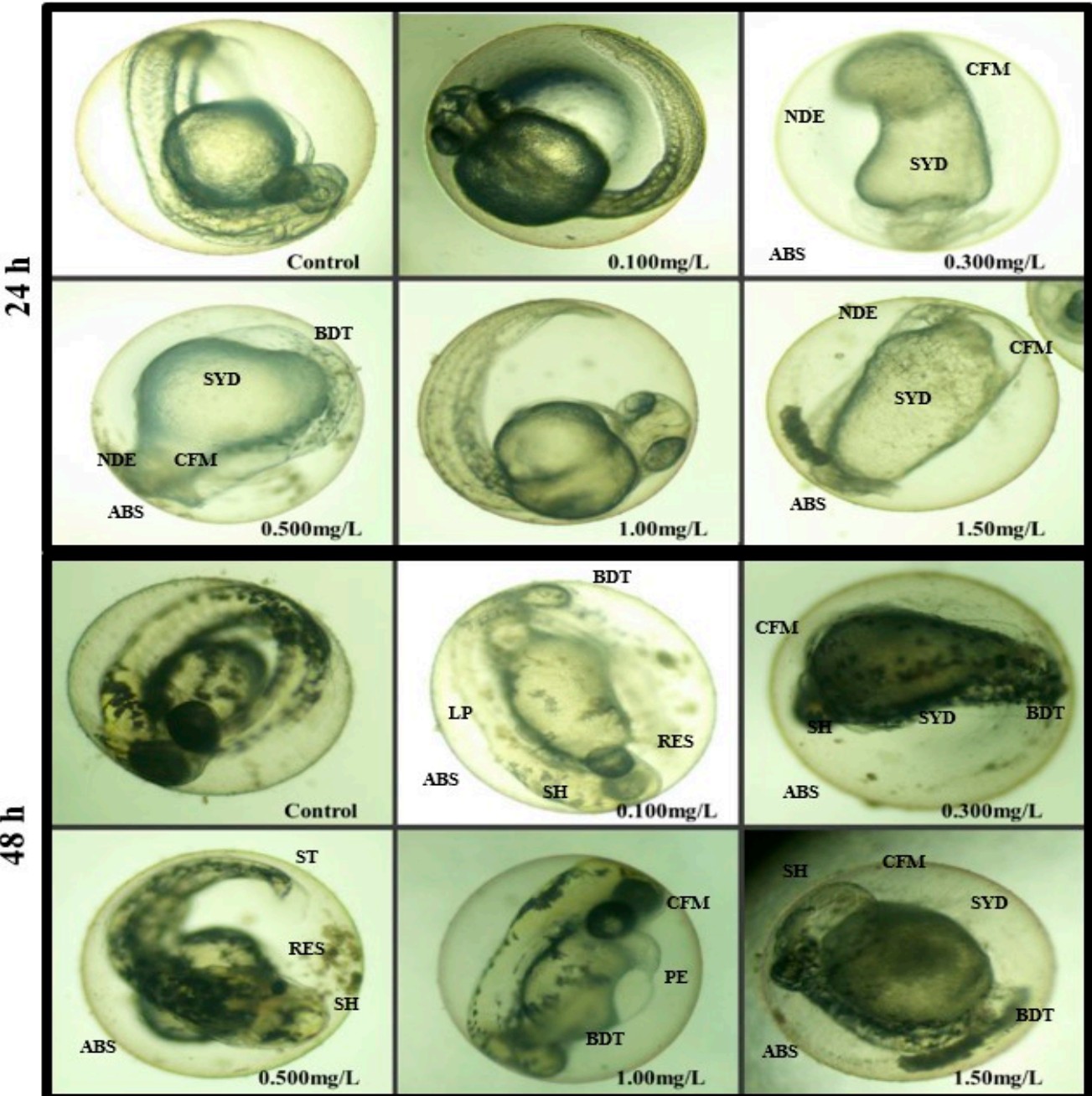

**Figure 2.** Zebrafish embryos were exposed to BP-3 for 24 or 48 h. Morphological changes were observed in embryos exposed to BP-3 from the first 24 h of exposure, including severe yolk deformation (SYD), lack of pigmentation (LP), reduced eye size (RES), abnormal body shape (ABS), non-development of eyes (NDE), small head size (SH), craniofacial malformation (CFM), barely detached tail (BDT), pericardial edema (PE), and short tail (ST). No morphological changes were found in the control group.

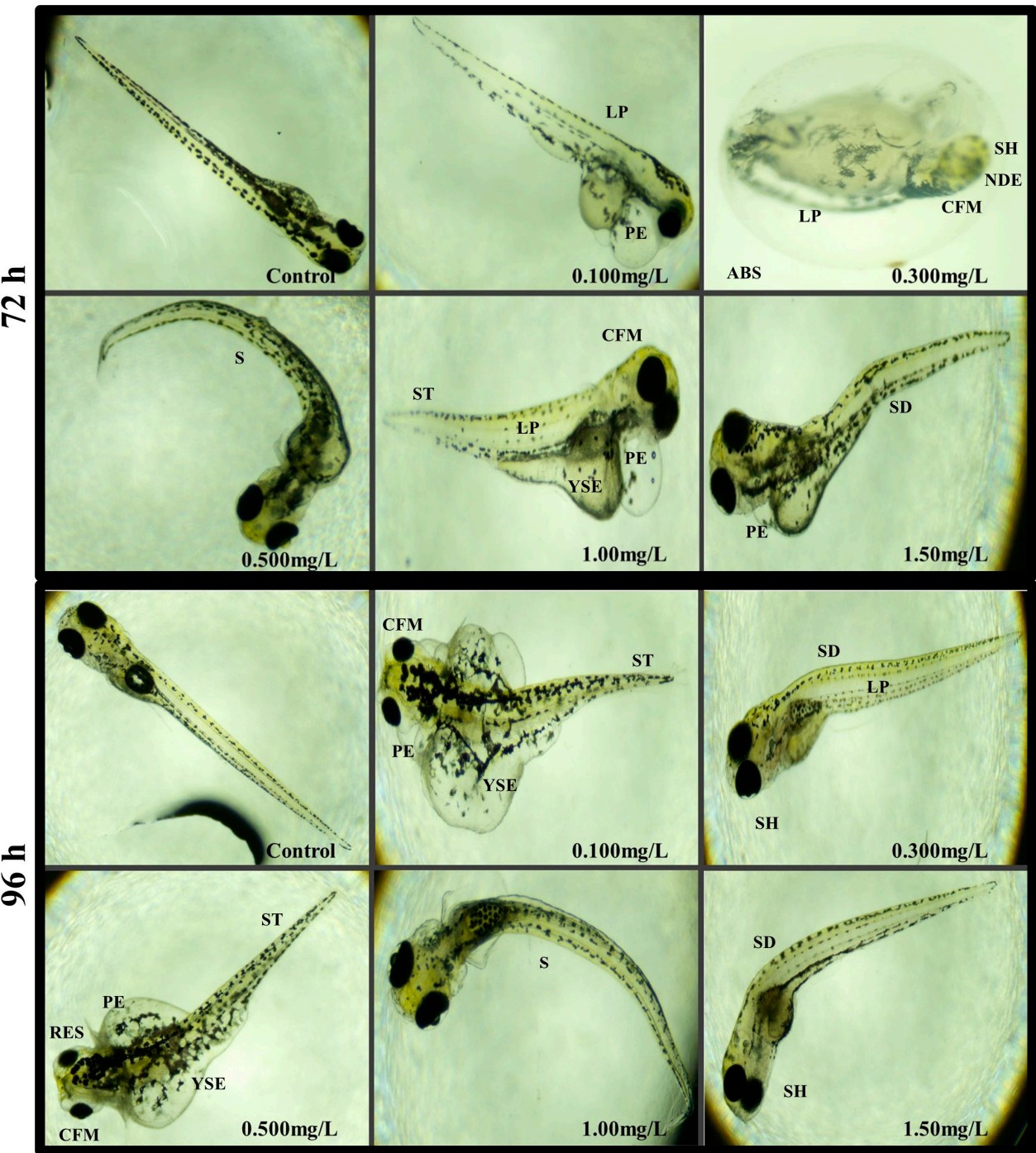

**Figure 3.** Zebrafish eleutheroembryos exposed to BP-3 for 72 and 96 h. Morphological changes in embryos exposed to BP-3 after hatching include pericardial edema (PE), spinal deformities (SD), scoliosis (S), yolk sac edema (YSE), lack of pigmentation (LP), reduced eye size (RES), abnormal body shape (ABS), non-development of eyes (NDE), small head (SH), craniofacial malformation (CFM), and short tail (ST). No morphological changes were found in the control group.

The other potential factor contributing to the observed morphological changes is endocrine disruption induced by BP-3 in organisms. Alterations caused by BP-3 are similar in adult males and eleutheroembryos, causing potential adverse effects on the endocrine system [7]. Exposure to chemicals capable of interfering with the endocrine system can lead to developmental malformations, disruptions in reproduction, increased cancer risk, and alterations in immune and nervous system function (EPA, 2024). Endocrine disruption

induced by BP-3 might result from its impact on gene expression in the AhR receptor, leading to a reduction in its expression within the pituitary gland [43]. The pituitary gland is divided into two main parts: the neurohypophysis (NH) and the adenohypophysis (AH). The AH is further categorized, with a focus here on category 1, which includes prolactin (PRL). In zebrafish, PRL primarily regulates salt and water homeostasis by altering ion retention and water absorption through peripheral osmoregulatory organs [44]. Additionally, growth hormone controls general metabolism and somatic growth [45]. Therefore, if somatic growth is affected, it can be inferred that larvae exposed to BP-3 may exhibit morphological changes.

In the first 24 h of exposure, dead embryos with coagulation were observed in all the treatments. At concentrations of 0.500 and 1.00 mg/L, embryo development seemed somewhat delayed compared to the other treatments. After 48 h of exposure, hatching began in all groups. In the control group, they began to swim without difficulty. At 96 h in all the concentrations investigated, most of the zebrafish larvae experienced locomotor issues, swimming circularly, and exhibiting slow swimming; they swam and went to the bottom motionless, while others remained motionless and had delayed responses to external stimuli. These effects could be attributed to the morphological changes observed in certain individuals following exposure to BP-3, or to some neurological effects. Zebrafish are an emerging alternative model for neurological studies of locomotion and behavior [46]. Another notable observation we wish to emphasize is that at concentrations of 0.500 mg/L, there was noticeable depigmentation in the larvae. This shift in pigmentation was observed when comparing the control and exposed groups. The embryonic pigment pattern is formed by cells derived from neural crest cells that migrate along specific pathways to their final position in the embryo [47]. These cells called melanophores contain a black pigment called melanin that forms the pigment pattern in zebrafish skin. These results suggest that melanin production is influenced by the presence of BP-3.

### 3.4. Quantification of Oxybenzone in Zebrafish Eleutheroembryos Tissue

To validate the method, five spikes were performed for analysis, with recovery percentages ranging from 98% to 107%. Spike and recovery analyses were employed to assess whether the matrix complexity of biological samples influenced analyte detection. The limit of quantification was 0.012 mg/L, and the limit of detection was 0.004 mg/L. No detectable BP-3 concentration was observed in the control group. Variations in eleutheroembryo numbers due to exposure-related mortality were noted across all groups. The wet weights of the exposed samples ranged from 0.0064 g to 1.1163 g, revealing significant differences among the treatments ($p < 0.05$). The highest BP-3 concentration ($0.74 \pm 0.13$ mg/L) was observed in the group exposed to 1.5 mg/L BP-3 (Figure 4). BP-3 has been detected at concentrations up to 123 ng/g lipid in fish [5]. Conversely, the group with the lowest BP-3 concentration was exposed to 0.1 mg/L, registering at $0.079 \pm 0.012$ mg/L. Notably, the group exposed to 0.1 mg/L exhibited the highest percentage of BP-3 absorption at $79 \pm 0.12\%$, while the group with the lowest absorption percentage was exposed to 0.50 mg/L, obtained $28 \pm 0.12\%$ (Figure 5).

Because the skin of embryos and young larvae is very thin (less than 10 μm thick) [48], xenobiotics can be transferred to their body more easily. Like other organic UV filters, BP-3 is a photostable, lipophilic, and potentially bio-accumulative compound. The relatively high log-Kow value of 3.8 implies slow biodegradation and an inclination to adsorb suspended solids and sediments [49], and they have been shown to accumulate in the food chain (bioaccumulation and potential biomagnification) [50]. Hence, it can be quantified in tissues within a short exposure period. It can be highlighted that BP-3 demonstrates substantial bioconcentration in zebrafish larvae, with concentrations ranging between those observed in environmental samples and higher. This phenomenon is attributed to its capacity to absorb up to 79% of the BP-3 concentration in water within a 96-h exposure period to zebrafish eleutheroembryos. These findings can be extrapolated to other oviparous aquatic

species, considering that approximately 90% of bony fish and 43% of cartilaginous fish exhibit oviparous reproduction [51].

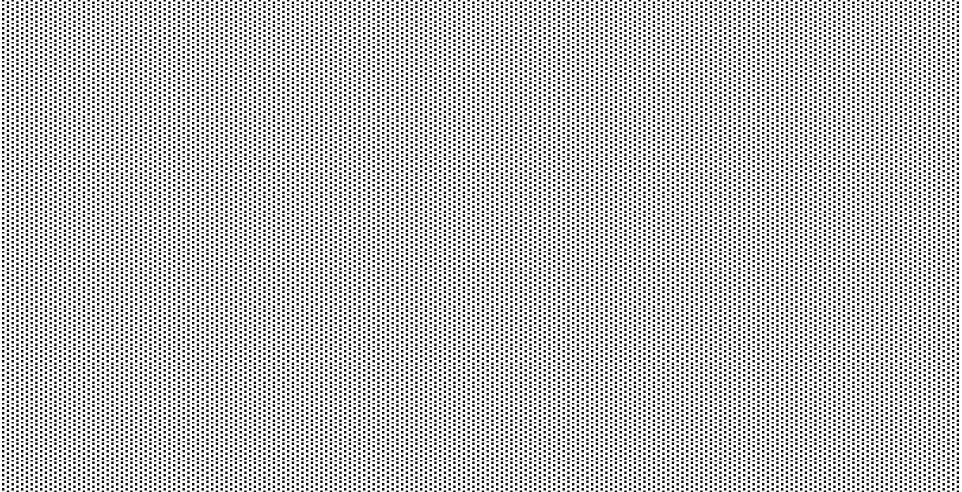

**Figure 4.** Oxybenzone concentration in zebrafish eleutheroembryonic tissue after 96 h exposure. Significant differences were found between the treatments ($p < 0.05$). The average is presented with the standard deviation ($n = 5$).

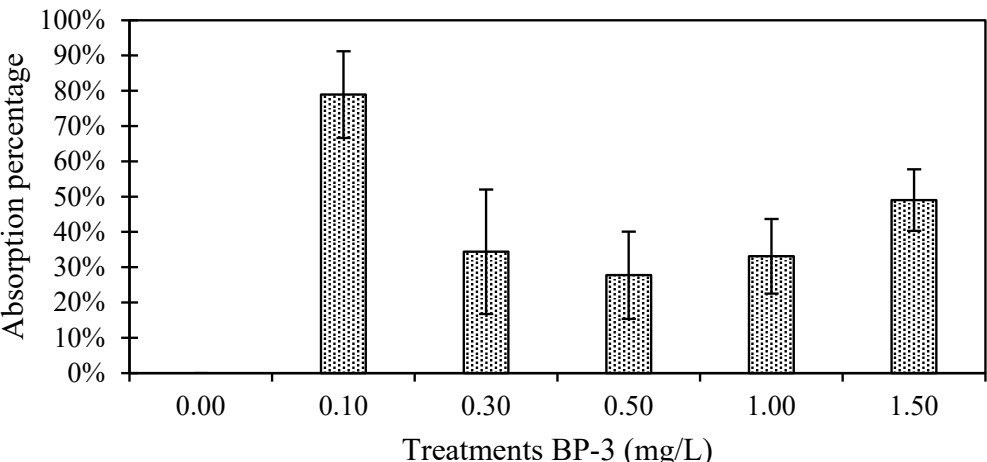

**Figure 5.** Uptake percentages in zebrafish eleutheroembryonic tissue after 96 h exposure. Significant differences were found between the treatments ($p < 0.05$). The average is presented with the standard deviation ($n = 5$).

The extraction method for BP-3 from the tissue presented here is an innovative approach that is not found in the current literature. This method was developed by applying the principles of green chemistry, emphasizing waste reduction, heightened energy efficiency, use of a singular solvent for extraction, and incorporation of microvolumes. Additionally, our methodology adhered to the principles of the 3Rs in biomedical research (Replacement, Reduction, and Refinement), employing zebrafish embryos as a small-scale model. Extractions were performed on samples containing a limited quantity of embryos, ranging from 25 to 45. This novel BP-3 extraction method is much simpler as it does not require the use of cartridges or expensive separation equipment, which typically results in the disposal of a significant amount of organic waste, as is the case with the most commonly used solid-phase extraction technique.

## 4. Conclusions

These findings suggest that BP-3 does not exert a significant impact on the mortality and hatching of zebrafish embryos during the 96-h exposure under the researched conditions. However, it does influence the development of both brine shrimp and zebrafish embryos and demonstrates absorption of the compound in zebrafish eleutheroembryo tissue. The rapid uptake of xenobiotics by tissues at low concentrations affirms the potential for bioaccumulation and biomagnification in natural environments when BP-3 is introduced. This raises concerns regarding its entry into the food chain, reaching humans through the consumption of fish or other aquatic organisms. The endocrine disrupting potential of this xenobiotic adds to the significance of this observation [52]. The identified morphological changes may render populations more vulnerable to premature death, potentially resulting from factors such as predation, reduced mobility for feeding, and/or organ failure. Conversely, *A. salina* exhibited an elevated mortality rate coupled with morphological changes, signifying heightened toxicity in this species. These findings have potentially detrimental effects on species and can affect population levels. Alterations in these populations may result in cascading effects on organisms at higher trophic levels. The presence of morphological changes across all concentrations suggested teratogenic effects in zebrafish embryos. A suggested approach is to investigate potential genetic alterations in embryos and *A. salina* exposed to BP-3. This exploration aims to elucidate whether the observed morphological changes stem from induced mutations. Further investigation is recommended to ascertain whether xenobiotics affect the development and formation of melanophores in zebrafish larvae. If we continue to use products that contain oxybenzone, more will reach water bodies. Environmental concentrations will continue to increase, affecting the organisms that live in aquatic ecosystems.

**Supplementary Materials:** The following supporting information can be downloaded at: https://www.mdpi.com/article/10.3390/jox14020032/s1, Figure S1: Percentage of physical malformations.

**Author Contributions:** All the Authors contributed to the design of the study. Conceptualization, methodology, validation, formal analysis, investigation, and the writing of the first manuscript draft was done by M.I.O.-R.; data collection by M.I.O.-R. and I.M.C.-M.; writing and review editing were performed by M.I.O.-R., I.M.C.-M. and F.R.R.-V. Resources, writing—review and editing and funding acquisition, F.R.R.-V. All authors have read and agreed to the published version of the manuscript.

**Funding:** This study was funded by the United States Department of Agriculture (USDA), award number NIFA-HSI 2020-38422-32258, NIFA RIIA 2020-70004-33081, and NIFA-AGFEI 2020-70004-32394, and by the National Science Foundation (NSF), award number HRD 1345156.

**Institutional Review Board Statement:** The study was conducted following the guidelines of the National Research Council Guide for the Care and Use of Laboratory Animals and the animal study protocol was approved by the Institutional Animal Care Use Committee (IACUC) of the University of Puerto Rico at Mayagüez (Office of Laboratory Welfare assurance number D20-01098) (3 October 2023).

**Informed Consent Statement:** Not applicable.

**Data Availability Statement:** The datasets generated during the study are available from the corresponding author upon request.

**Acknowledgments:** In honor of Oscar J. Perales Perez from the University of Puerto Rico at Mayagüez. We extend our sincere gratitude to agronomist Alan J. Figueroa Ruiz for generously providing a microscope equipped with a camera, which greatly facilitated the capture of images depicting the development of zebrafish embryos and *Artemia salina*. I extend my appreciation to Chadeline Reyes Martinez, Adriana S. Torres Rodríguez, Beatriz M. Purcell Collazo, and Cristian C. Pagán Vega for their dedicated support in the laboratory.

**Conflicts of Interest:** The authors declare no conflicts of interest.

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
