# Peer review of "Toxicity of UV Filter Benzophenone-3 in Brine Shrimp Nauplii (Artemia salina) and Zebrafish (Danio rerio) Embryos"

_jox, doi:10.3390/jox14020032_

Round 1
Reviewer 1 Report
Comments and Suggestions for Authors
The manuscript describes some basic studies evaluating the effects of BP-3 on artemia and zebrafish. The presented studies add to existing knowledge on BP-3 effects in wildlife. However, there are many weak issues that must be addressed prior to a possible publication. A rationale should be given, as to why this study was necessary and a hypothesis should be presented. The presentation of the data should largely be improved. There is a serious question on the statistics (data must be related to the solvent control). Unfortunately, the studies on morphological effects are purely descriptive with no quantification and with no possible explanation on the effects noted. Better conclusions should be formulated.
Specific comments are as follows:
Please be consistent in the use of ”BP-3” and ”oxybenzone” throughout the manuscript.
Please check the spacing throughout the manuscript. Place a space between the number and the unit of measurement, e.g. change “0.600mg/L” to ”0.600 mg/L”. Delete double spacing and other unintentional spacing. Insert spacing where needed.
Line 2: Please replace “3-Benzophenone” with “Benzophenone-3”.
Line 17 and 18: Please use italics for “A. salina” and “D. rerio”.
Line 42: Please provide a reference for this statement.
Line 51-53: Please add “OECD test guideline 236 [24]”
Line 92-94: Please add that the 1.5% MeOH corresponds to the methanol concentration in the 3.0 mg/L exposure group and that the 2.5% MeOH corresponds to the methanol concentration in the 5.0 mg/L exposure group.
Line 101: Please delete the comma.
Line 103: Please replace “Reverse Osmosis/ UV water” with “Reverse osmosis (RO)+UV water”.
Line 122: Please replace “reverse osmosis (RO)/UV water” with ”RO+UV water”.
Line 147-148: “and subsequent analyses were not performed.” What is meant by this statement?
Line 145-149: Unclear. In which order were these steps performed? Was the water not removed before weighing and freezing?
Line 185: Please replace ”Method” with ”method”.
Line 191-193 and Table 2: Three different 48h LC50 values are given! Please sort this out.
Line 194-195: Please rephrase to: ”There were no deaths in the synthetic sea-water control group (Fig. 1).” and move this line to line 211 (before: ”In the solvent…”)
Line 195-198: This reviewer cannot follow these arguments. ”The results obtained for LC50 can be compared with those of A. franciscana, where the LC50 value at 48 h was 5.27 mg/L [21], which is very close to our result at 24 h of exposure in A. salina. This observation suggested that A. salina may exhibit greater resistance to xenobiotics.” Please explain.
Figure 1: Please also include the mortality of the solvent control groups.
Line 203: Please change “Salina” to ”salina”.
Line 204-205 + 258-259 + 327-328 + 331-332 : This does not make sense: ”The mean is presented as the standard deviation”. Please rephrase.
Line 206-211 + Figure 2: Some of the pictures are of very poor quality and the morphological changes in the intestinal tract are not described. If no better pictures or quantification of the changes exist, the parts about morphological changes in artemia should be omitted from the manuscript: Figure 2, Figure 3 and line 206-211.
Line 211-213: Was the same mortality rate observed in both solvent control groups (1.5 and 2.5% methanol)? Otherwise the mortality rates for each solvent control group and time point should be given and discussed accordingly.
Line 212+216: Please change MeOH to methanol.
Line 211-221: These claims need to be substantiated by statistics. The study is flawed by the use of different solvent concentrations in the different treatment groups.
Line 239-240: Please rephrase this sentence.
Line 245-246: This applies at 25°C.
Line 249-252: These sentences seem out of context: “However, a significant effect on development was observed. According to our results, exposure to concentrations exceeding 0.100 mg/L of BP-3 could harm these fish, as well as other organisms.” Please move or delete.
Section 3.3 + Figure 4 + Figure 5: Some of the pictures are of rather poor quality. Quantification of and statistics on the morphological changes should be provided.
Line 273+277: Please include year in the reference: Incardona et al.
Line 313: What do “spikes” mean in this context? Spike-and-recovery controls? Please explain.
Line 326: Change ”exposition” to ”exposure”.
Line 339-341: This sentence is unclear: ”We can infer that BP-3 is a markedly bioaccumulative compound at environmental concentrations, as it can absorb up to 79% of its concentration in the water within 96 h of contact with zebrafish eleutheroembryos.”. Please rephrase.
Line 373-375: This sentence is unclear: ”An in-depth exploration of the mechanism by which BP-3 influences the development of both organisms is suggested, leading to observed morphological changes.”. Please rephrase.
References:
Please write all authors instead of ”et al.”
Line 423-426: Reference 12 and 13 are duplicates.
Line 424: Please correct “ans” to “and”
Line 443: Please correct “themarinemicroalga” to ” the marine microalga”.
Line 452-455: Reference 24 and 25 are duplicates.
Line 458: Some info is missing for reference 27.
Line 459: Author info is missing for reference 28.
Table 1:
The choice of references included in the table seems a bit random. Consider deleting the study with gilt-head bream, and include missing references dealing with the effects of BP-3 in zebrafish, including:
Bai et al., 2023: Lifetime exposure to benzophenone-3 at an environmentally relevant concentration leads to female–biased social behavior and cognition deficits in zebrafish.
Tao et al., 2023: Embryonic benzophenone‑3 exposure inhibited fertility in later‑life female zebrafish and altered developmental morphology in offspring embryos.
Wang et al 2023: Comparison of developmental toxicity of benzophenone-3 and its metabolite benzophenone-8 in zebrafish.
Moreira & Luchiari 2022: Effects of oxybenzone on zebrafish behavior and cognition.
Sun et al., 2021: Influence of organic colloids on the uptake, accumulation and effects of benzophenone-3 in aquatic animals.
Xu et al., 2021: Effects of Low Concentration Benzophenone-3 Exposure on the Sex Ratio and Offspring Development of Zebrafish (Danio rerio).
Meng et al., 2020: Toxic effects and transcriptome analyses of zebrafish (Danio rerio) larvae exposed to benzophenones.
Tao et al., 2020: Environmental relevant concentrations of benzophenone-3 induced developmental neurotoxicity in zebrafish.
Lee et al., 2018: Thyroid Hormone-Disrupting Potentials of Major Benzophenones in Two Cell Lines (GH3 and FRTL-5) and Embryo-Larval Zebrafish.
Kinnberg et al., 2015: Endocrine-disrupting effect of the ultraviolet filter benzophenone-3 in zebrafish, Danio rerio.
Please change the table caption as the table does not contain “studies of the use of BP-3”.
Please consider deleting the solvent and “Material and” from the table and just include BP-3 concentrations for all studies.
Please delete “/authors” and “Balázs et al., (2016)”
Please change “concentra-tions” to “concentrations”
Please change ”Embryo water” to ”embryo water”, “Condition factor” to “condition factor”, “Salina” to ”salina”, ”Nauplii Instar” to ”nauplii instar”, ”Brine Shrimp” to ”brine shrimp” and ”Marine Microalga” to ”marine microalga”
In conclusion, although the morphological effects are interesting, if the presentation of the morphological effects cannot be significantly improved and supported by quantification and statistical evaluations, this reviewer would strongly recommend changing this manuscript into a short communication focusing on mortality and quantification of BP-3 in zebrafish.
Comments on the Quality of English Language
The language is somewhat casual and unclear. Please see "Suggestions for Authors" for more specific comments.
Reviewer 2 Report
Comments and Suggestions for Authors
The study explores oxybenzone (BP-3), a prevalent UV filter in personal care products, and its effects on zebrafish embryos and Artemia salina. Findings suggest BP-3 influences the development of both organisms, posing potential threats to aquatic ecosystems. Although informative, the article exhibits deficiencies that require attention. Consequently, I propose a minor revision to rectify the identified issues.
- The manuscript contains typographical errors such as missing commas, dots, and italics. Additionally, certain abbreviations were used inconsistently or with errors. For example, on line 47, "OECD" should be used. Also, on lines 270 and 279, “yolk sac edema” should be abbreviated as "YSE". Furthermore, there are instances of missing or double spaces (e.g., line 320, 365). Please ensure consistency and correctness (e.g., “...xenobiotics can be transferred to THEIR body more easily…”).
- Please review the references and make necessary adjustments in the manuscript accordingly. For instance, ensure that duplicate references such as 12 and 13, as well as 24 and 25, are rectified. Additionally, correct other references such as reference 33 “20199”.
- Corrections are needed in Table 1, including fixing the spelling (e.g., "concentra-tions") and providing an explanation of the SPE method.
- It is essential to add a paragraph detailing reagents, references, and commercial sources. This information should be removed from subsequent sections to enhance clarity and conciseness.
- Increase the font size in Schemes 1 and 2 for better readability.
- In Figure 1, ensure that the numbers on the axis correspond accurately to the bars for clarity and precision of data representation.
- Further explanation is required for certain statements. For instance, elaborate on how oxybenzone induces an erratic swimming pattern in brine shrimp (line 207).
- Sections containing lines 206-221 and lines 231-237 require clearer organization. For example, focus first on elucidating the effects of the solvent used, and then delve into explaining the experimental results to enhance readability and comprehension.
- Provide a brief explanation of the Reed-Muench [26] and Probit [27] methods utilized to calculate the lethal concentration.
- Expand discussions in Sections 3.2 and 3.3 to provide a more comprehensive analysis. Apart from comparing the results with findings from other studies, elucidate similarities and discrepancies, and seek explanations.
- It is recommended to conduct in vitro studies before proceeding with experiments on zebrafish and shrimp. Please provide precedents.
- Given the potential for increased usage leading to higher concentrations of BP-3 over time, it may be reasonable to test concentrations exceeding those typically encountered in natural waters. Additionally, exploring higher concentrations of BP-3 could provide valuable insights into its ecological impact, considering the potential for bioaccumulation.
Round 2
Reviewer 1 Report
Comments and Suggestions for Authors
Although the manuscript has been improved, major concerns regarding the morphological changes still remain.
Specific comments:
This reviewer does not understand Response 11, Line 211-221 and repeats that the claims need to be substantiated by statistics, and that the study is flawed by the use of different solvent concentrations in the different treatment groups.
This reviewer still recommends omitting the parts about morphological changes in artemia including Figure 2 and Figure 3 from the manuscript since some of the pictures are still of very poor quality and the morphological changes in the intestinal tract are still not described and no statistics are provided.
Line 239-240 (new: 254-255) has not been rephrased.
Section 3.3 + Figure 4 + Figure 5: Some of the pictures are still of rather poor quality. Since no quantification and no statistics on the morphological changes are provided, the value of the descriptive morphological changes can be questioned.
This reviewer does not understand Response 11, Line 245-246. The claim applies at 25°C. The authors used a water temperature of 27 °C ± 1 °C. Please address this.
Please correct reference 13.
Table 1:
The choice of references included in the table seems random. Please address this. Consider deleting the study with gilt-head bream, and include missing references dealing with the effects of BP-3 in zebrafish, including:
Bai et al., 2023: Lifetime exposure to benzophenone-3 at an environmentally relevant concentration leads to female–biased social behavior and cognition deficits in zebrafish.
Tao et al., 2023: Embryonic benzophenone‑3 exposure inhibited fertility in later‑life female zebrafish and altered developmental morphology in offspring embryos.
Wang et al 2023: Comparison of developmental toxicity of benzophenone-3 and its metabolite benzophenone-8 in zebrafish.
Moreira & Luchiari 2022: Effects of oxybenzone on zebrafish behavior and cognition.
Sun et al., 2021: Influence of organic colloids on the uptake, accumulation and effects of benzophenone-3 in aquatic animals.
Xu et al., 2021: Effects of Low Concentration Benzophenone-3 Exposure on the Sex Ratio and Offspring Development of Zebrafish (Danio rerio).
Meng et al., 2020: Toxic effects and transcriptome analyses of zebrafish (Danio rerio) larvae exposed to benzophenones.
Tao et al., 2020: Environmental relevant concentrations of benzophenone-3 induced developmental neurotoxicity in zebrafish.
Lee et al., 2018: Thyroid Hormone-Disrupting Potentials of Major Benzophenones in Two Cell Lines (GH3 and FRTL-5) and Embryo-Larval Zebrafish.
Kinnberg et al., 2015: Endocrine-disrupting effect of the ultraviolet filter benzophenone-3 in zebrafish, Danio rerio.
In conclusion, although the morphological effects are interesting, since the presentation of these effects cannot be significantly improved and supported by quantification and statistical evaluations, this reviewer would strongly recommend changing this manuscript into a short communication by omitting the morphological effects and focusing only on the results on mortality and quantification of BP-3 in zebrafish.
Comments on the Quality of English LanguageThe language is somewhat casual and unclear.
Round 3
Reviewer 1 Report
Comments and Suggestions for Authors
Please consider simplifying the information in Table 1 by deleting solvent information and only including the most important and relevant results from each study.
Please include a figure for the percentage of physical malformations in artemia similar to the figure for zebrafish (Fig. S. 1).
Line 238: Don’t you mean “one side” of the body instead of “one end”?
Line 260-261: This sentence has still not been rephrased.
Line 267-268: It should still be addressed that this solubility applies at 25°C, which is below the temperature used for zebrafish.
Comments on the Quality of English LanguageThe language is somewhat casual and unclear.
